# Potential Epha2 Receptor Blockers Involved in Cerebral Malaria from *Taraxacum officinale*, *Tinospora cordifolia*, *Rosmarinus officinalis* and *Ocimum basilicum*: A Computational Approach

**DOI:** 10.3390/pathogens11111296

**Published:** 2022-11-04

**Authors:** Mohd Sayeed Shaikh, Fahadul Islam, Parag P. Gargote, Rutuja R. Gaikwad, Kalpana C. Dhupe, Sharuk L. Khan, Falak A. Siddiqui, Ganesh G. Tapadiya, Syed Sarfaraz Ali, Abhijit Dey, Talha Bin Emran

**Affiliations:** 1Shreeyash Institute of Pharmaceutical Education and Research, Aurangabad 431136, India; 2Department of Pharmacy, Faculty of Allied Health Sciences, Daffodil International University, Dhaka 1341, Bangladesh; 3Department of Pharmaceutical Chemistry, N.B.S. Institute of Pharmacy, Ausa 413520, India; 4Sub District Hospital Ambad, Jalna 431204, India; 5Department of Life Sciences, Presidency University, 86/1 College Street, Kolkata 700073, India; 6Department of Pharmacy, BGC Trust University Bangladesh, Chittagong 4381, Bangladesh

**Keywords:** cerebral malaria, EphA2 receptor, *Tinospora cordifolia*, *Taraxacum officinale*, *Rosmarinus officinalis*, docking

## Abstract

Cerebral malaria (CM) is a severe manifestation of parasite infection caused by *Plasmodium* species. In 2018, there were approximately 228 million malaria cases worldwide, resulting in about 405,000 deaths. Survivors of CM may live with lifelong post-CM consequences apart from an increased risk of childhood neurodisability. EphA2 receptors have been linked to several neurological disorders and have a vital role in the CM-associated breakdown of the blood–brain barrier. Molecular docking (MD) studies of phytochemicals from *Taraxacum officinale*, *Tinospora cordifolia*, *Rosmarinus officinalis*, *Ocimum basilicum*, and the native ligand ephrin-A were conducted to identify the potential blockers of the EphA2 receptor. The software program Autodock Vina 1.1.2 in PyRx-Virtual Screening Tool and BIOVIA Discovery Studio visualizer was used for this MD study. The present work showed that blocking the EphA2 receptor by these phytochemicals prevents endothelial cell apoptosis by averting ephrin-A ligand-expressing CD8+ T cell bioadhesion. These phytochemicals showed excellent docking scores and binding affinity, demonstrating hydrogen bond, electrostatic, Pi-sigma, and pi alkyl hydrophobic binding interactions when compared with native ligands at the EphA2 receptor. The comparative MD study using two PDB IDs showed that isocolumbin, carnosol, luteolin, and taraxasterol have better binding affinities (viz. −9.3, −9.0, −9.5, and −9.2 kcal/mol, respectively). *Ocimum basilicum* phytochemicals showed a lower docking score but more binding interactions than native ligands at the EphA2 receptor for both PDB IDs. This suggests that these phytochemicals may serve as potential drug candidates in the management of CM. We consider that the present MD study provides leads in drug development by targeting the EphA2 receptor in managing CM. The approach is innovative because a role for EphA2 receptors in CM has never been highlighted.

## 1. Introduction

Cerebral malaria (CM) is a severe manifestation of a parasitic infection caused by the *Plasmodium* species. *P. falciparum* and *P. vivax* are the species responsible for most of the complicated forms of CM in humans. In 2018, there were an estimated approximately 228 million cases of malaria worldwide, resulting in about 405,000 deaths [1]. Approximately 20% of children admitted to the hospital with CM have died [2]. Of these, 67% were children under the age of 5 years [1,3]. Patients that survive CM have life-long post-CM consequences and an increased risk of developing neurological and cognitive deficits, behavioral difficulties, and epilepsy, therefore making CM a leading cause of childhood neurodisability [1,3,4]. Plasmodium falciparum erythrocyte membrane protein-1 (PfEMP1) protein is synthesized during the parasite’s erythrocytic schizogony stage inside the RBC [5]. PfEMP1 acts as both an antigen and an adhesion protein [3]. PfEMP1 protein is then expressed on the RBC membrane. Receptors for this protein are also found in the endothelial cells of blood vessels and healthy RBCs [6,7]. Thus, the *P. falciparum*-infected RBCs (iRBCs) bind with the endothelial cells and healthy RBCs. There is an activation of immune response by the production of antibodies, which in turn leads to the release of inflammatory cytokines (LT-α and TNF-α), chemokines (CXCL10 and CCL2), ROS and RNS, all of which injure the brain tissues [8,9]. The binding of antibodies to PfEMP1 disables the binding properties of its DBL domains, thus resulting in a loss of cell adhesion, and the iRBC is destroyed; thereby, CM is prevented [9]. However, to escape the host’s immune response, different *P. falciparum* switch on and off different *var* genes to produce antigenically distinct PfEMP1s [6,7]. Each variant type of PfEMP1 has different binding properties and therefore is not recognized by the human immune system’s antibodies every time [3].

Disruption of the blood–brain barrier (BBB) is the key feature of CM, which may cause complications such as seizures and coma. Increased barrier permeability occurs due to structural disruption of adherence junctions present between the endothelial cells of the BBB [10,11]. It is clear that this barrier can be disrupted by two mechanisms: (a) apoptosis of endothelial cells and (b) opening of the tight junctions [11]. An experimental mouse model of cerebral malaria (ECM) has demonstrated that the T cells play a crucial role in the development of this condition [12]. Previous work suggests that cytotoxic T cells accumulate in the brain in response to inflammation induced by appropriated PfEMP1 protein iRBCs [12]. EphA2 receptors have been linked to several neurological disorders and have a major role in CM due to their association with the breakdown of BBB [13,14]. Here, we represent that EphA2 is a critical target protein facilitating the endothelial cell apoptosis process by targeting ephrin-A ligand-expressing CD8+ T cell adhesion [13]. This, along with the binding of soluble ephrin-A ligands, initiates signaling pathways, which in turn induces the opening of the tight junctions between the endothelial cells of the BBB. The purpose of the present work is to develop an adjunct therapy for CM based on blocking the EphA2 receptor. Analysis of the ephrin-A ligand’s soluble protein form and the form expressed on peripheral CD8+ T cells, both of which bind on the EphA2 receptor, has a strong correlation with the pathogenesis of CM. The crucial assumption is that EphA2 is upregulated on brain endothelial cells post-inflammation process, and it mediates the adhesion of ephrin-A ligand expressed CD8+ T cells, which in turn facilitates the process of degranulation and endothelial apoptosis [13]. EphA2-mediated signaling pathways will be triggered by the binding of CD8+ T cell-bound and the soluble form of ephrin-A ligands, and this will, in turn, mediate the opening of the tight junctions in the BBB [13]. The rationale for the present work is that the EphA2 receptor could serve as a novel target for the development of an adjunct therapy for CM. To date, there is no treatment protocol for CM that focuses on the EphA2 receptor as a target.

Molecular docking (MD) reveals various types of interactions of ligands with target EphA2 receptors, specifically via hydrogen and electrostatic bonds. The purpose of this research was to discover the chemical constituents from medicinal plants *Taraxacum officinale* (Dandelion), *Tinospora cordifolia* (Guduchi), *Rosmarinus officinalis* (Rosemary) and *Ocimum basilicum* (Basil) as potential blockers of EphA2 receptor through the MD study. Finding an effective therapy for CM is the need of the hour. Novel medicines/vaccines from sources apart from traditional or herbal remedies requires time-consuming clinical trials and long-term approval procedures. In contrast, we could develop a cure from traditional or herbal medicines relatively quickly [15]. Here, we have investigated some phytochemicals from these plants through an MD study to serve this purpose in managing CM.

## 2. Materials and Methods

### 2.1. Molecular Docking

To study the interactions between phytochemicals and the EphA2 receptor, MD experiments of phytochemicals derived from *Taraxacum officinale*, *Tinospora cordifolia*, *Rosmarinus officinalis*, and *Ocimum basilicum*, together with the natural ligand ephrin-A, were conducted. The Autodock Vina 1.1.2 in PyRx-Virtual Screening Tool 0.8 software of the Chimera version 1.10.2 and the BIOVIA Discovery Studio Visualizer (version 19.1.0.18287) were used for the MD study [16]. All the docking poses, ligand, and protein interactions were studied by Discovery studio software which enables us to identify the types of interactions.

### 2.2. Ligand Preparation

In this study, the phytoconstituents of *Taraxacum officinale*, *Tinospora cordifolia*, *Rosmarinus officinalis*, *Ocimum basilicum*, and native ligand ephrin-A (SDF file) were obtained from the official website of the US National library of medicine PubChem (http://pubchem.ncbi.nlm.nih.gov/, accessed on 5 June 2022). Then, using the open babel tool, structures were imported into PyRx-Virtual Screening Tool 0.8 software, and the energy minimization process was performed by considering fundamental parameters based on the elements and their hybridization by a universal force field (UFF) [17]. The discovery studio software was employed for the prediction of the active sites of the selected EphA2 receptor.

### 2.3. Target Preparation

For the MD study, a three-dimensional grid box (size_x = 26.3786A°; size_y = 29.0004A°; size_z = 20.8096A°) was designed to define the area for interactions in the occupied cavity of EphA2receptor using Autodock tool 1.5.6 with exhaustiveness value of 8. The cavity was defined with the assistance of the Toggle Selection Spheres option that was provided in the Vina Wizard Tool of PyRx 0.8. This option was used to choose the active amino acid residues. The grid box was precisely positioned so that it could occupy all of the active binding sites as well as the critical residues. All the phytochemicals and EphA2 receptors were then subjected to docking to obtain the possible affinities/interactions with each other. The complete molecular docking procedure was performed as described by S. L. Khan et al. [17]. The complete molecular crystal structure of EphA2 Receptor Protein Kinase with PDB ID—6FNH and 5NK0 has been utilized MD study, which was developed by Kudlinzki D., Troester A. et al. (2018) and Kudlinzki D., Linhard V. L. et al. (2017).

## 3. Results and Discussion

### 3.1. Development of CM Associated with EphA2

The breakdown of the blood–brain barrier occurs during the blood stage of iRBCs in the schizont stage of *Plasmodium* infection. These iRBCs travel through the bloodstream and adhere to various receptors that are expressed on brain microvascular endothelial cells. These receptors include EPCR, ICAM-1, and other unknown receptors [5,6,13]. It has been demonstrated that ephrin-A ligand expression in the circulation (in soluble form as well as CD8+ T cell surface bound form) is elevated in *P. falciparum*-infected children with symptoms of CM (Figure 1) [13]. The approach is innovative because the role of EphA2 receptors in CM has not been highlighted until now. The present work is significant because the identification of new targets for adjunct therapies in CM is urgently needed.

### 3.2. Prediction of ADME Parameters

We have used the SwissADME (http://www.swissadme.ch, accessed on 10 June 2022) for the estimation of in silico ADME parameters of all our phytoconstituents. This provides insights into their pharmacokinetic behavior. To ensure drug-like properties, Lipinski’s rule of five is a prerequisite for rational drug design. All of the phytoconstituents under our study meet the criteria of Lipinski’s rule of five (mol. wt. ≤ 500 Da; log P o/w ≤ 5; HBD ≤ 5; HBA ≤ 10; Solubility (LogS): ≥ 4). It was found that all our phytoconstituents have the values of the ADME parameters in the requisite range, and hence, they possess drug-like characteristics as per Lipinski’s rule of five [18].

Estimation of the pharmacokinetic parameters of drug molecules enables researchers to predict some of their important biological aspects. In order to predict whether or not the compounds in question are optimum for oral bioavailability, Lipinski’s rule of five and Veber’s rules were utilized. All phytoconstituents were studied for their ADMET characteristics to assess their pharmacokinetic profiles and drug-likeness (Table 1).

We have investigated phytoconstituents from the *Taraxacum officinale*, *Tinospora cordifolia*, *Rosmarinus officinalis*, and *Ocimum basilicum* so as to identify the potential lead molecules through molecular docking study of their binding specificity in the target receptor cavity and binding free energies for the therapeutics management of CM. In compliance with Lipinski’s and Veber’s rules (Table 2), all of these phytoconstituents have demonstrated the characteristics of drug-like molecules, and also no violation of both of these rules by any phytoconstituent was observed (except taraxasterol, which violates one parameter viz., MLOGP > 4.15) [19]. All phytoconstituents have calculated log *p* values within the required range of the Lipinski rule of 5. The obtained values indicate good lipophilicity and high GI absorption along with access to the brain by crossing the BBB. Hence it is concluded that these phytoconstituents can probably be potential herbal lead compounds for the management of CM, especially in children. These phytoconstituents have an excellent binding affinity towards the EphA2 receptor, as predicted by the values obtained through molecular docking studies, and thus, they can act as EphA2 blockers. The Log *p* values of these phytoconstituents show the fair permeability of these drugs in the body to enter the target site in CM viz. the brain, and their capacity to bind with the EphA2 receptor. All these phytoconstituents were found to have the required values of mol. wt. ≤ 500 Da; log P o/w ≤ 5; HBD ≤ 5; HBA ≤ 10, and thus they have not violated the Lipinski rule of 5 (except taraxasterol). Additionally, these phytoconstituents have not violated the criterion as per Veber’s rule (viz. total polar surface area values, i.e., TPSA ≤ 140 and the number of rotatable bonds ≤ 10), and the values comply as they fall within the acceptable range for oral bioavailability [19].

It is concluded from the obtained values that the phytoconstituents under the present study show a good BBB penetration potential. Thus, they can be targeted for delivery to the central nervous system and may serve as potential lead compounds for the pharmacotherapy of CM. These phytoconstituents exhibit optimum log Kp (skin permeation, cm/s) and bioavailability scores and are readily permeated to the brain. All the phytoconstituents under study have values in the acceptable range of Ghose, Egan, and Muegge filters (Table 1). These phytocompounds display high lipophilicity and good GI absorption, and also they do not violate the Lipinski rule and Veber’s rule [18].

### 3.3. In Silico Toxicity Prediction Study Employing ProTox-II Toxicity Explorer

The in silico toxicity risk study of phytoconstituents using the open-source program ProTox-II toxicity explorer (https://tox-new.charite.de/protox_II/, accessed on 12 June 2022) (Table 3) was performed, which revealed that all our phytoconstituents were nontoxic [18]. Thus, they serve as promising leads for the therapeutic management of CM as adjuvant therapy in children. Some of the phytoconstituents viz., Methyl Chavicool, Palmatine, Magnoflorine, Carnosol, Rosmarinic Acid, Taraxasterol, and Taraxinic Acid show the biologic response of immunotoxicity. This actually might be helpful in the prevention of the neuroinflammatory immune response in the pathogenesis of CM, as neurodegeneration in CM is associated with the neuroinflammatory response which ultimately leads to the disruption of the BBB. As immunotoxicity for our phytoconstituents is predicted based on the structural database library in the software, it needs to be proved through pharmacological screening and toxicity study. It was also revealed that all of our phytoconstituents are inactive at the nuclear receptors, such as the Aromatase receptor, Androgen Receptor (AR), and Estrogen Receptor Alpha (ER). An exception to this generalization is the phytoconstituent Luteolin obtained from the *Taraxacum officinale* showed Estrogen Receptor Alpha (ER) active effects. ProTox-II also predicts LD50 values, and it was found that all of our phytoconstituents have significant LD50 scores, which indicates their safety and nontoxicity [18].

### 3.4. Docking Studies Using PyRx

EphA2 belongs to the family of tyrosine kinase receptors expressed mostly in hepatocytes, brain, and other tissues that plays important roles in tissue organization, homeostasis, and various pathological processes [20]. The ephA2 receptor is activated in response to binding with ephrin protein. EphA2 has a crucial role in CM associated with the breakdown of BBB integrity via neuroinflammatory responses, immune cell activation, platelets activation, and increased oxidative stress that all together leads to apoptosis of endothelial cells [9,11,12,13]. To prevent these episodes of CM, the plants under the present investigation have reported effects that can abolish such complications. A molecular docking study has shown that the blocking of the EphA2 receptor by phytochemicals from *Taraxacum officinale*, *Tinospora cordifolia*, *Rosmarinus officinalis*, and *Ocimum basilicum* prevents endothelial cell apoptosis by averting ephrin-A ligand-expressing CD8+ T cell bioadhesion, as shown in Figure 2.

*Taraxacum officinale* has been used in traditional medicine in Europe, North America, and China. The phytochemicals of this plant have various biological activities such as hepatoprotective, anti-inflammatory, anti-diabetic, immune modulation, and anti-rheumatic [21,22]. *Tinospora cordifolia* is considered an essential herb in Ayurveda. It has pharmacologically proven effects such as antioxidant, antimicrobial, anti-malarial, antibacterial, and antifungal [23,24]. It is also utilized in treating hepatic and renal dysfunction [25]. *Rosmarinus officinalis* is considered one of the sacred plants to ancient Egyptians, Romans, and Greeks, and it has pharmacologically authenticated medicinal activities such as antioxidant, anti-inflammatory, hepatoprotective, antiulcer, anticancer, antiviral, antimicrobial, antiproliferative, improving cognitive deficits, neuroprotective and many more [26]. Rosmarinic acid has proven activities such as antioxidant, anti-inflammatory, control of hypercholesterolemia, oxidative stress and mental fatigue, and reduced lipid peroxidation in the heart and brain [27]. Similarly, antiangiogenic and neuroprotective effects are observed for carnosic acid (benzenediol abietane diterpene) and carnosol (phenolic diterpene) [28]. Essential oils from *Ocimum basilicum* have important therapeutic roles such as inflammatory, anticancer, antioxidant, stomachache, antimicrobial, antiviral, larvicidal and antileishmanial, and anti-aging [29]. They can prevent oxidative stress in these disease conditions and are thus used to treat cardiovascular diseases and jaundice [30]. Docking of phytochemicals from these plants on the *EphA2* receptor with 6FNH and 5NK0 was performed, as reported in Table 4 and Table 5. All the phytochemicals were successfully docked on the EphA2 receptor having 6FNH and 5NK0. PubChem ID, molecular formula, molecular weight (gm/mol), binding affinity (kcal/mol), hydrogen bonds, and active amino acid residues for the said receptor are shown in Table 4 and Table 5. 3D and 2D images of docking poses showing the chemical structure of ligands and phytochemicals with the EphA2 receptor, which enables us to predict groups that evolve in interaction with EphA2, are shown in Figure 3 and Figure 4.

It is observed that all the phytochemicals of these plants have more binding affinity and interactions than the native ligands of both PDB IDs, except those of phytochemicals from *Ocimum basilicum*, as shown in Table 4 and Table 5 (Linalool, Methyl Eugenol, Methyl Chavicool). These phytochemicals from *Ocimum basilicum* have a lower molecular weight than that of native ligands, which might be the probable reason for their lower docking scores (binding affinity). Although these phytoconstituents demonstrate a lower docking score (binding affinity), they have greater binding interactions (viz. hydrogen bond, electrostatic, and Pi-sigma and pi alkyl hydrophobic bindings) as compared to the native ligand at the EphA2 receptor for both PDB IDs, as shown in Figure 3 and Figure 4. Luteolin and isocolumbin have shown better binding affinity values of −9.5 and −9.3 kcal/mol than the native ligand (PubChem ID: 134693866) on the EphA2 receptor (6FNH), respectively. Luteolin has shown more binding interactions with 6 hydrogen bonds with bond lengths 2.67, 2.26, 2.54, 2.22, 2.41, 3.54 A° with the active amino acid residues ASN744, THR692, GLU623, MET695, ASN744, respectively, as shown in Table 4 and Figure 3. It has also demonstrated 8 hydrophobic bonds with Pi-sigma and pi-alkyl type hydrophobic bonds with bond lengths 3.92, 5.35, 4.43, 4.74, 4.46, 5.49, 5.14, 4.51 A° showing the interactions with active amino acid residues VAL627, ILE619, ALA644, LEU746, ALA644, LX5646, LX5646, LEU746, ILE619, respectively. Isocolumbin shows less binding interactions and more binding affinity viz. −9.3 kcal/mol as compared to the native ligand (PubChem ID: 134693866). It demonstrates two hydrogen bonds having bond lengths 2.55 and 3.73 A° with GLN848 and PHE604, respectively, and one alkyl hydrophobic interaction with bond length 2.43 A° with ARG860 residue. The docking analysis has also shown that Taraxinic Acid (Dandelion) has a lesser binding affinity (−6.8 kcal/mol) and a single alkyl hydrophobic interaction having bond length 4.32 A° with LEU746 residue on EphA2 receptor (6FNH).

The docking study on the EphA2 receptor (5NK0) has shown that Taraxasterol and Isocolumbin showed the highest binding affinity (−9.2 and −9.0 kcal/mol) than native ligand (−7.6 kcal/mol) as shown in Table 4 and Figure 4 (PubChem ID: 127053578). Results from a docking study on EphA2 with 5NK0 have shown that Taraxinic Acid has more binding affinity (−8.7 kcal/mol) and increased binding interactions as compared to the molecular structure with 6FNH. Taraxinic Acid (5NK0) shows the 4-hydrogen bonding with bond lengths 2.45, 2.60, 2.98, 2.87 Aº with residues H-O, GLU696, LYS646, LYS646, and 5 hydrophobic interactions with bond length 4.34, 3.66, 4.24, 4.55, 5.24 with active amino acid residue VAL627, ALA644, LYS646, VAL627, ALA644 at the target receptor EphA2, as represented in Table 5 and Figure 4. The docking study on the EphA2 receptor (5NK0) reveals that Isocolumbin, Carnosol, Rosmarinic Acid, Carnosic Acid, and Taraxasterol have higher binding affinity values and lesser binding interactions as compared to the native ligand (PubChem ID: 127053578). MD study revealed that the blockade of EphA2 by phytochemicals from *Taraxacum officinale*, *Tinospora cordifolia*, *Rosmarinus officinalis*, and *Ocimum basilicum* could prevent endothelial cell apoptosis by averting ephrin-A ligand-expressing CD8+ T cell bioadhesion.

## 4. Conclusions

EphA2 is identified as a new drug target in the host for the specific treatment of CM. Present in silico molecular docking study using phytochemicals of four plants under investigation viz., *Taraxacum officinale*, *Tinospora cordifolia*, and *Rosmarinus officinalis* have shown an encouraging result by blockade of the EphA2 receptor.

The present study has revealed that Palmatine, Magnoflorine, Isocolumbin, Carnosol, Rosmarinic Acid, and Carnosic Acid are the phytoconstituents from these plants that have a very good binding affinity score and favorable binding interactions with EphA2. The docking study of phytochemicals Isocolumbin, Carnosol, and Luteolin with EphA2 (6FNH) revealed the best binding affinity scores −9.3, −9.0, and −9.5 kcal/mol, respectively. While in the case of molecular docking with EphA2 (6FNH) phytochemicals Isocolumbin, Carnosol, Taraxasterol, and Luteolin have the highest binding affinity 8.2, 7.9, 9.2, and 9.0 kcal/mol, respectively. However, phytochemicals from *Ocimum basilicum* (basil) have a lesser binding affinity but undergo good binding interactions with the active residues of the EphA2 receptor. It is observed that Isocolumbin, Carnosol, Taraxasterol, and Luteolin have good binding affinities and target interactions, indicating the stability of the ligand receptor complex formed. This indicates that they are potential candidates to be used as a drug against CM.

## Figures and Tables

**Figure 1 pathogens-11-01296-f001:**
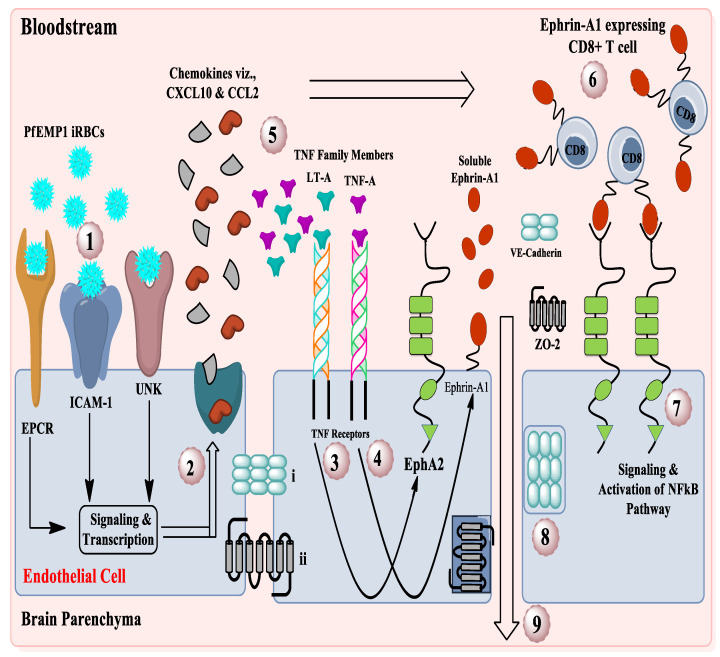
EphA2-associated cellular biochemical mechanism of development of CM. (1) iRBCs directed signaling through EPCR, ICAM-1, and UNK receptors lead to endothelial activation. (2) Release of various pro-inflammatory cytokines (LT-α and TNF-α and chemokine (CXCL10 and CCL2). (3) The cytokine LT-α can act on proximal endothelial cells to induce upregulation of the receptor EphA2. (4) TNF-α induces upregulation of ephrin-A1 ligand, which can be cleaved by metalloproteinases and is released into the bloodstream. (5) Chemokines such as CXCL10 and CCL2 recruit circulating immune cells, including CD8+ T cells, to the brain to the site of inflammation. (7) Ephrin-A1 ligand is then adhered to newly recruited CD8+ T cells and considered as ephrin-A1 ligand expressing CD8+ T cells. (7) Upon entry into the brain microvasculature, CD8+ T cells expressing the ephrin-A1 ligand bind to the EphA2 receptor expressed on brain endothelial cells leading to clustering and activation of EphA2. Forward signaling cascades from the EphA2 receptor led to the activation of the NF-κB pathway. (8) This results in various downstream consequences, including disruption of endothelial cell junctions due to both internalization and shedding of different adherents and tight junction protein components. (8) Once brain endothelial cell junctions are disrupted, contents of the vasculature can leak into the brain parenchyma. (9) This leads to vascular leakage, brain edema, and the development of other neurological symptoms associated with *P. falciparum* infection in CM (Modified diagram of Darling et al. 2020) [13].

**Figure 2 pathogens-11-01296-f002:**
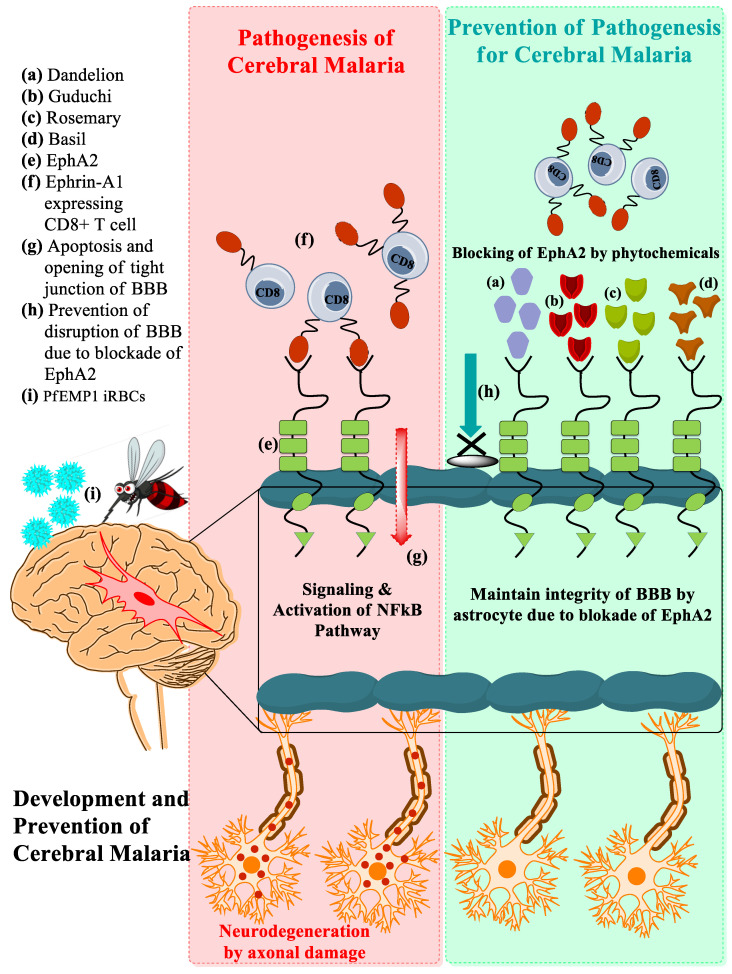
Pathogenesis of development and prevention of CM.

**Figure 3 pathogens-11-01296-f003:**
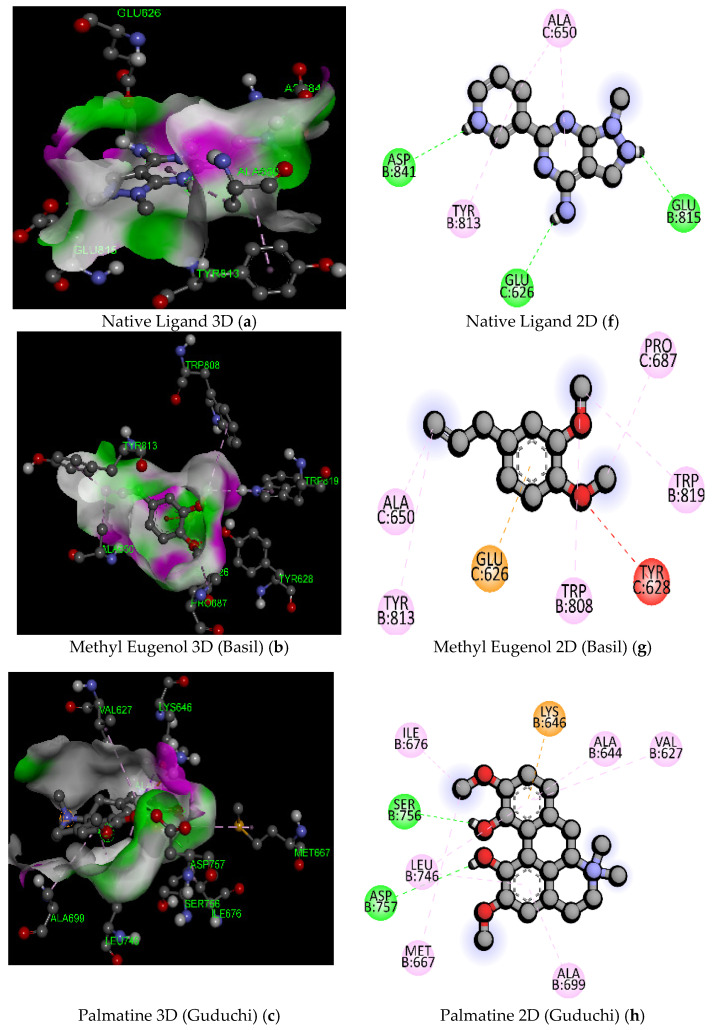
2D and 3D docking poses of native ligand (**a**,**f**) and phytoconstituents [Methyl Eugenol (**b**,**g**); Palmatine (**c**,**h**); Carnosic Acid (**d**,**i**); Luteolin (**e**,**j**)] with Receptor-binding Domain (RBD) of EphA2 (PDB ID: 6FNH).

**Figure 4 pathogens-11-01296-f004:**
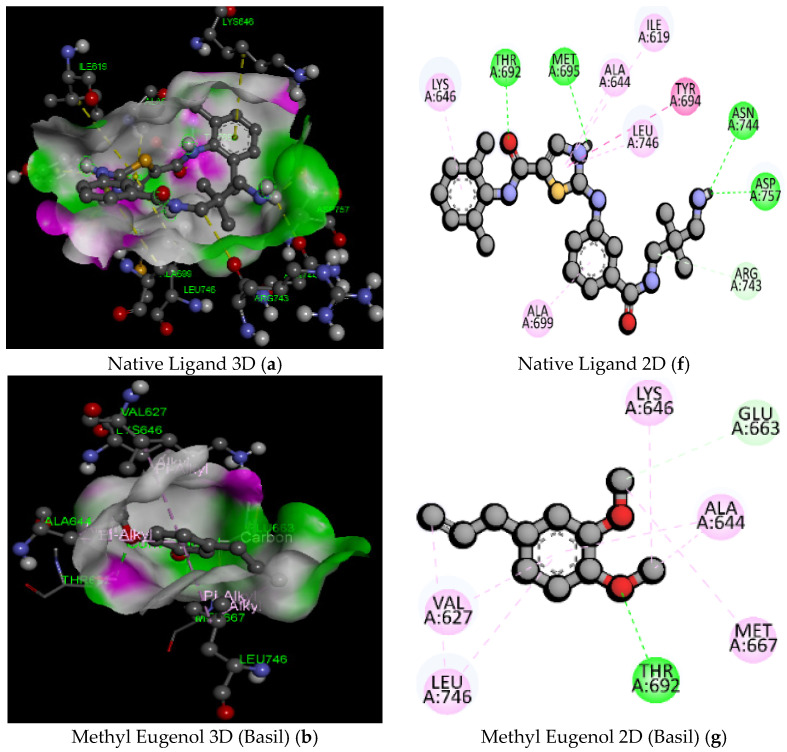
2D and 3D docking poses of the native ligand (**a**,**f**) and phytochemicals [Methyl Eugenol (**b**,**g**); Palmatine (**c**,**h**); Carnosic Acid (**d**,**i**); Luteolin (**e**,**j**)] with Receptor-binding Domain (RBD) of EphA2 (PDB ID: 5NK0).

**Table 1 pathogens-11-01296-t001:** Pharmacokinetic and drug-likeness properties of phytoconstituents.

Sl. No.	Parameter and Compound Name	GI * Absorption	BBB * Permeability	P. gp* Substrate	CYP1A2 Inhibitor	CYP219 Inhibitor	CYP2C9 Inhibitor	CYP2D6 Inhibitor	CYP3A4 Inhibitor	Log Kp(Skin Permeation	Ghose	Egan	Muegge	Bioavailability
1	Linalool	High	Yes	No	No	No	No	No	No	−5.13 cm/gm	01 MW < 160	Yes	2 MW < 2000, heteroatoms < 2	0.55
2	Methyl Eugenol	High	Yes	No	Yes	No	No	No	No	−5.60 cm/s	Yes	Yes	No	0.55
3	Methyl Chavicool (Basil)	High	Yes	No	Yes	Yes	No	No	No	−5.34 cm/s	Yes	Yes	Yes	0.55
4	Palmatine	High	Yes	Yes	Yes	No	No	Yes	Yes	−5.79 cm/s	Yes	Yes	yes	0.55
5	Magnoflorine	High	Yes	Yes	No	No	No	Yes	Yes	−6.44 cm/s	Yes	Yes	yes	0.55
6	Isocolumbin	High	No	Yes	No	No	No	No	No	−6.95 cm/s	Yes	Yes	Yes	0.55
7	Carnosol	High	Yes	Yes	No	No	Yes	No	Yes	−5.01 cm/s	Yes	Yes	Yes	0.55
8	Rosmarinic Acid	Low	No	No	No	No	No	No	No	−6.82 cm/s	Yes	1 TPSA > 131.6	Yes	0.58
9	Carnosic Acid	High	No	No	No	No	Yes	No	No	−4.86 cm/s	Yes	Yes	Yes	0.56
10	Taraxasterol	Low	No	NO	NO	No	No	No	No	−2.42 cm/s	3 WLOGP > 5.6, MR > 130, atoms > 70	1 WLOGP > 5.88	2 XLOGP3 > 5, heteroatoms < 2	0.55
11														
	Luteolin (Dandelion)	High	No	No	Yes	No	No	Yes	Yes	−6.25 cm/s	Yes	Yes	Yes	0.55
12	Taraxinic Acid (Dandelion)	High	yes	NO	NO	No	NO	NO	No	−6.89 cm/s	Yes	Yes	Yes	0.85

* GI-gastrointestinal; BBB-blood–brain barrier; P-gp-p-glycoprotein.

**Table 2 pathogens-11-01296-t002:** Molecular formula and drug-likeness properties of phytoconstituents.

Sr. No.	Compounds	Molecular Formula	Lipinski Rule of 5					Veber’s Rule	
			Molecular Weight	HBA *	HBD *	Log P	Violation	Total Polar Surface Area (TPSA) (A^2^)	Number of Rotatable Bonds
1	Linalool (Basil)	C10H18O	154.25	1	1	2.66	00	20.23	4
2	Methyl Eugenol (Basil)	C11H14O2	178.23	2	0	2.58	00	18.48	4
3	Methyl Chavicool (Basil)	C11H12O3	192.21	3	0	2.30	00	35.53	4
4	Palmatine (Guduchi)	C21H22NO4+	352.40	4	0	2.53	00	41.85	4
5	Magnoflorine (Guduchi)	C20H24NO4+	342.41	4	2	1.88	00	62.16	2
6	Isocolumbin (Guduchi)	C20H22O6	358.4	6	1	2.13	00	85.97	1
7	Carnosol (Rosemary)	C21H28O4	344.44	4	2	4.03	00	66.76	1
8	Rosmarinic Acid (Rosemary)	C18H16O8	360.31	8	5	1.52	00	144.52	7
9	Carnosic Acid (Rosemary)	C20H28O4	332.43	4	3	3.82	00	77.76	2
10	Taraxasterol (dandelion)	C30H50O	426.72	1	1	7.11	01, MLOGP > 4.15	20.23	0
11	Luteolin (Dandelion)	C15H10O6	286.24	6	4	1.73	00	111.13	1
12	Taraxinic Acid (Dandelion)	C15H18O4	262.30	4	1	2.12	00	63.60	1

* HBA—Hydrogen bond acceptor; HBD—Hydrogen bond donor.

**Table 3 pathogens-11-01296-t003:** Toxicity assessment and nuclear receptor signaling activation of marketed drug.

Sl. No	Compound	Predicted LD50 Value (mg/Kg)/Toxicity Class	Hepatotoxicity	Carcinogenicity	Immunotoxicity	Cytotoxicity	Nuclear Receptor Signaling Pathways Active		
							Androgen Receptor (AR)	Aromatase Active	Estrogen Receptor Alpha (ER)
1	Linalool (Basil)	2200/5	None	None	None	None	None	None	None
2	Methyl Eugenol (Basil)	810/4	None	Yes	None	None	None	None	None
3	Methyl Chavicool (Basil)	7900/6	None	None	Yes	None	None	None	None
4	Palmatine (Guduchi)	200/3	None	None	Yes	None	None	None	None
5	Magnoflorine (Guduchi)	401/4	None	None	Yes	None	None	None	None
6	Isocolumbin (Guduchi)	280/3	None	None	None	None	None	None	None
7	Carnosol (Rosemary)	287/3	None	None	Yes	None	None	None	None
8	Rosmarinic Acid (Rosemary)	5000/5	None	None	Yes	None	None	None	None
9	Carnosic Acid (Rosemary)	287/3	None	None	None	None	None	None	None
10	Taraxasterol (Dandelion)	5000/5	None	None	Yes	None	None	None	None
11	Luteolin (Dandelion)	3919/5	None	None	None	None	None	None	Yes
12	Taraxinic Acid (Dandelion)	900/4	None	None	Yes	None	None	None	None

**Table 4 pathogens-11-01296-t004:** MD study of phytoconstituents with EphA2 (PDB ID: 6FNH) representing the binding affinities (kcal/mol), hydrogen bonds, and active amino acid residues with their bond length (A°).

Sl. No.	Compound Name	Molecular Formula/Molecular Weight (gm/mol)	Docking Score/Binding Affinity (kcal/mol)	Active Amino Acid Residue	Bond Length (A°)	Bond Category	Bond Types
1	Native Ligand (DXK) (PubChem ID: 134693866)	C_11_ H_10_ N_6_/226.24	−7.3	GLU815	2.17	Hydrogen Bond	Conventional Hydrogen Bond
				ASP841	2.65		
				GLU626	2.28		
				ALA650	3.94	Hydrophobic	Alkyl
				ALA650	4.69		Pi-Alkyl
				TYR813	5.27		
2	Linalool (Basil)	C_10_H_11_O/154.25	−4.9	VAL627	5.03	Hydrophobic	Alkyl
				LEU746	4.81		
				MET695	5.25		
				LEU746	4.14		
				ALA644	3.99		
3	Methyl Eugenol (Basil)	C_11_H_14_O_2_/178.23	−5.6	GLU626	3.40	Electrostatic	Pi-Anion
				ALA650	3.86	Hydrophobic	Alkyl
				PRO687	4.01	Hydrophobic	Alkyl
				TRP808	4.62	Hydrophobic	Pi-Alkyl
				TYR813	4.95	Hydrophobic	Pi-Alkyl
				TRP819	4.77	Hydrophobic	Pi-Alkyl
4	Methyl Chavicool (Basil)	C_10_H_12_O/148.20	−5.4	GLU815	3.77	Hydrogen Bond	Carbon Hydrogen Bond
				GLU626	3.53	Electrostatic	Pi-Anion
				ALA650	4.08	Hydrophobic	Alkyl
				PRO687	4.10	Hydrophobic	Alkyl
				MET840	5.46	Hydrophobic	Pi-Alkyl
				TYR813	5.04	Hydrophobic	Pi-Alkyl
				TYR628	5.41	Hydrophobic	Pi-Alkyl
5	Palmatine (Guduchi)	C_21_H_22_NO_4_^+^/352.4	−8.4	GLN855	2.43	Hydrogen Bond	Conventional Hydrogen Bond
				MET733	5.02	Hydrophobic	Alkyl
				MET851	4.26	Hydrophobic	Alkyl
				MET851	5.21	Hydrophobic	Pi-Alkyl
				PHE604	4.57	Hydrophobic	Pi-Alkyl
6	Magnoflorine (Guduchi)	C_20_H_24_NO_4_^+^/342.4	−7.9	ASP757	2.75	Hydrogen Bond	Conventional Hydrogen Bond
				SER756	2.00	Hydrogen Bond	Conventional Hydrogen Bond
				LYS646	4.63	Electrostatic	Pi-Cation
				LYS646	3.90	Hydrophobic	Pi-Sigma
				MET667	4.74	Hydrophobic	Alkyl
				ILE676	3.64	Hydrophobic	Alkyl
				ALA699	5.16	Hydrophobic	Pi-Alkyl
				LEU746	4.89	Hydrophobic	Pi-Alkyl
				VAL627	5.47	Hydrophobic	Pi-Alkyl
				ALA644	5.08	Hydrophobic	Pi-Alkyl
				LEU746	5.18	Hydrophobic	Pi-Alkyl
7	Isocolumbin (Guduchi)	C_20_H_22_O_6_/358.4	−9.3	GLN848	2.55	Hydrogen Bond	Conventional Hydrogen Bond
				PHE604	3.73	Hydrogen Bond	Carbon Hydrogen Bond
				ARG860	5.19	Hydrophobic	Alkyl
8	Carnosol (Rosemary)	C_20_H_26_O_4_/330.40	−9.0	GLN669	2.43	Hydrogen Bond	Conventional Hydrogen Bond
				GLN855	2.43	Hydrogen Bond	Conventional Hydrogen Bond
				GLN852	3.03	Hydrogen Bond	Conventional Hydrogen Bond
				GLN855	2.26	Hydrogen Bond	Conventional Hydrogen Bond
				SER671	3.57	Hydrogen Bond	Carbon Hydrogen Bond
9	Rosmarinic Acid (Rosemary)	C_18_H_16_O_8_/360.3	−7.6	SER756	2.54	Hydrogen Bond	Conventional Hydrogen Bond
				TYR694	3.25	Hydrogen Bond	Pi-Donor Hydrogen Bond
				LYS646	3.80	Hydrophobic	Pi-Sigma
				N:UNK1	5.27	Hydrophobic	Pi-Pi Stacked
				ALA644	5.30	Hydrophobic	Pi-Alkyl
				LEU746	5.27	Hydrophobic	Pi-Alkyl
10	Carnosic Acid (Rosemary)	C_20_H_28_O_4_/332.4	−7.3	THR605	2.20	Hydrogen Bond	Conventional Hydrogen Bond
				LYS603	2.35	Hydrogen Bond	Conventional Hydrogen Bond
				GLN848	2.58	Hydrogen Bond	Conventional Hydrogen Bond
				PHE604	3.06	Hydrogen Bond	Carbon Hydrogen Bond
				ILE870	5.41	Hydrophobic	Alkyl
11	Taraxasterol (Dandelion)	C_30_H_50_O/426.7	−8.9	ARG743	3.66	Hydrogen Bond	Carbon Hydrogen Bond
				VAL627	4.89	Hydrophobic	Alkyl
				ALA644	4.40	Hydrophobic	Alkyl
				LEU746	4.68	Hydrophobic	Alkyl
12	Luteolin (Dandelion)	C_21_H_20_O_11_/448.4	−9.5	ASN744	2.67	Hydrogen Bond	Conventional Hydrogen Bond
				H-O	2.26	Hydrogen Bond	Conventional Hydrogen Bond
				THR692	2.54	Hydrogen Bond	Conventional Hydrogen Bond
				GLU623	2.22	Hydrogen Bond	Conventional Hydrogen Bond
				MET695	2.41	Hydrogen Bond	Conventional Hydrogen Bond
				ASN744	3.54	Hydrogen Bond	Carbon Hydrogen Bond
				VAL627	3.92	Hydrophobic	Pi-Sigma
				ILE619	5.35	Hydrophobic	Pi-Alkyl
				ALA644	4.43	Hydrophobic	Pi-Alkyl
				LEU746	4.74	Hydrophobic	Pi-Alkyl
				ALA644	4.46	Hydrophobic	Pi-Alkyl
				LYS646	5.49	Hydrophobic	Pi-Alkyl
				LEU746	5.14	Hydrophobic	Pi-Alkyl
				ILE619	4.51	Hydrophobic	Pi-Alkyl
13	Taraxinic Acid (Dandelion)	C_21_H_28_O_9_/424.4	−6.8	LEU746	4.32	Hydrophobic	Alkyl

**Table 5 pathogens-11-01296-t005:** MD study of phytoconstituents with EphA2 (PDB ID: 5NK0) representing the binding affinities (kcal/mol), hydrogen bonds, and active amino acid residues with their bond length (A°).

Sl. No.	Compound Name	Molecular Formula/Molecular Weight (gm/mol)	Docking Score/Binding Affinity (kcal/mol)	Active Amino Acid Residue	Bond Length (A°)	Bond Category	Bond Types
1	Native Ligand (91E) (PubChem ID: 127053578)	C_23_H_26_ClN_5_O_2_S/472.00	−7.6	ASN744	2.83	Hydrogen Bond	Conventional Hydrogen Bond
				ASP757	2.15		
				MET695	2.15		
				THR692	2.00		
				ARG743	3.38		Carbon Hydrogen Bond
				TYR694	5.54	Hydrophobic	Pi-Pi Stacked
				ALA699	5.03		Pi-Alkyl
				ILE619	5.40		
				ALA644	4.15		
				LEU746	4.56		
				LYS646	4.76		
2	Linalool (Basil)	C_10_H_11_O/154.25	−5.1	LEU746	4.78	Hydrophobic	Alkyl
				ILE619	4.53		
				LEU746	5.25		
				ALA644	4.34		
				MET695	5.26		
				LEU746	4.38		
				ALA644	4.01		
				LYS627	4.44		
				VAL627	4.03		
				ALA644	4.88		
				TYR694	4.97	Hydrophobic	Pi-Alkyl
3	MethylEugenol (Basil)	C_11_H_14_O_2_/178.23	−5.2	THR692	2.44	Hydrogen Bond	Conventional Hydrogen Bond
				GLU663	3.73		Carbon Hydrogen Bond
				LEU746	4.02	Hydrophobic	Alkyl
				MET667	5.06		
				ALA644	3.76		
				LYS646	4.45		
				VAL627	5.41		Pi-Alkyl
				ALA644	5.07		
				LEU746	5.22		
4	Methyl Chavicool (Basil)	C_10_H_12_O/148.20	−5.1	LYS646	4.49	Hydrophobic	Alkyl
				MET667	5.15		
				ILE690	3.73		
				VAL627	5.23		
				LEU746	4.45		
				VAL627	4.76		Pi-Alkyl
				ALA644	4.61		
				LYS646	5.06		
5	Palmatine (Guduchi)	C_21_H_22_NO_4_^+^/352.4	−7.9	THR692	2.61	Hydrogen Bond	Conventional Hydrogen Bond
				ALA699	2.27		
				ILE690	3.70		Carbon Hydrogen Bond
				LEU746	3.57	Hydrophobic	Pi-Sigma
				ALA699	3.77		Alkyl
				ALA644	4.02		
				LYS646	4.30		
				LYS646	4.57		
				MET667	5.02		
				ILE690	3.77		
				ALA699	4.66		Pi-Alkyl
6	Magnoflorine (Guduchi)	C_20_H_24_NO_4_^+^/342.4	−7.8	H-O	1.90	Hydrogen Bond	Conventional Hydrogen Bond
				ARG743	3.76		Carbon Hydrogen Bond
				ASP757	3.68		
				TYR694	3.71		
				GLY698	3.36		
				ILE619	3.97	Hydrophobic	Pi-Sigma
				ILE619	3.98		Alkyl
				VAL627	5.17		Pi-Alkyl
				LEU746	5.28		
				ALA644	4.63		
				LEU746	4.58		
				TYR694	4.89		
7	Isocolumbin (Guduchi)	C_20_H_22_O_6_/358.4	−8.2	GLU663	2.52	Hydrogen Bond	Conventional Hydrogen Bond
				LYS646	2.55		
				ILE619	3.61	Hydrophobic	Pi-Sigma
8	Carnosol (Rosemary)	C_20_H_26_O_4_/330.40	−7.9	MET667	2.82	Hydrogen Bond	Conventional Hydrogen Bond
				LYS646	5.21	Hydrophobic	Pi- Alkyl
9	Rosmarinic Acid (Rosemary)	C_18_H_16_O_8_/360.3	−7.6	ILE690	2.12	Hydrogen Bond	Conventional Hydrogen Bond
				ASP757	3.81	Electrostatic	Pi-Anion
				LEU746	3.66	Hydrophobic	Pi-Sigma
				ASP757	3.93		
				LEU746	5.15		Pi-Alkyl
				VAL627	5.25		
				ALA644	3.83		
10	Carnosic Acid (Rosemary)	C_20_H_28_O_4_/332.4	−7.7	LEU746	3.70	Hydrophobic	Pi-Sigma
				LEU746	4.73	Hydrophobic	Alkyl
11	Taraxasterol (Dandelion)	C_30_H_50_O/426.7	−9.2	THR692	2.01	Hydrogen Bond	Conventional Hydrogen Bond
				VAL627	4.80	Hydrophobic	Alkyl
				ALA644	5.37		
				LYS646	4.22		
12	Luteolin (Dandelion)	C_21_H_20_O_11_/448.4	−9	H-O	1.90	Hydrogen Bond	Conventional Hydrogen Bond
				SER756	2.18		
				LYS646	2.82		
				THR692	2.48		
				TYR694	3.24		Pi-Donor Hydrogen Bond
				ILE619	3.75	Hydrophobic	Pi-Sigma
				LEU746	3.63		
				VAL627	5.04		Pi-Alkyl
				LEU746	5.41		
				ALA644	4.00		
				VAL627	5.43		
13	Taraxinic Acid (Dandelion)	C_21_H_28_O_9_/424.4	−8.7	H-O	2.45	Hydrogen Bond	Conventional Hydrogen Bond
				GLU696	2.60		
				LYS646	2.98		
				LYS646	2.87		
				VAL627	4.34	Hydrophobic	Alkyl
				ALA644	3.66		
				LYS646	4.24		
				VAL627	4.55		
				ALA644	5.24		

## Data Availability

Not applicable.

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
