# Peer review of "Potential Epha2 Receptor Blockers Involved in Cerebral Malaria from Taraxacum officinale, Tinospora cordifolia, Rosmarinus officinalis and Ocimum basilicum: A Computational Approach"

_pathogens, 2022, doi:10.3390/pathogens11111296_

Round 1
Reviewer 1 Report
Major Revisons:
The author has reported research paper entitled "Discovery of Potential blocker of the EphA2 Receptor: Involved in Cerebral Malaria from Tinospora cordifolia, Taraxacum officinale & Rosmarinus officinalis L. Through in-silico Study" which represent the current need in the cerebral malaria treatment. Some comments are presented below for revised manuscript:
- This manuscript is clearly written and contains some sections that are not well described.
- For example, the material and method part is not divided into sections and no significant background information about toxicity study and ADME.
- Aim of the study need to write at the last paragraph of Introduction section.
- The author provides Figures 3 and 4 in proper formation as per the journal’s specifications.
- Author should provide software utilized for the generation of Figure 1 and 2.
- Why authors choose Autodock Vina?
- The overall analysis and arrangement of this manuscript looks perfect.
- If authors decided to capitalize the compounds names, then it should be maintained throughout the entire manuscript.
- English languages need to improve carefully.
Author Response
- This manuscript is clearly written and contains some sections that are not well described.
Response: We thank the reviewer for the comments. All the sections of the manuscript have been thoroughly revised, and the corrections are highlighted.
- For example, the material and method part is not divided into sections and no significant background information about toxicity study and ADME
Response: Thanks for the comments and suggestions. There is no significant methodology adapted for toxicity and ADME studies. Because these studies were carried out by using open online sources. Hence methodology is not described. But the details of online web source in the form of web link is given in the results and discussion section. These parts are highlighted.
- Aim of the study need to write at the last paragraph of Introduction section.
Response: We thank the reviewer for the comments. The study's aim is written in the last paragraph of the Introduction section in the manuscript, and the same is highlighted.
- The author provides Figures 3 and 4 in proper formation as per the journal’s specifications.
Response: Thanks for the comments and suggestions. The proper formatting in Tables 3 and 4 is done as per the journal’s specifications.
- Author should provide software utilized for the generation of Figure 1 and 2.
Response: We thank the reviewer for the comments. Art made for Figures 1, and 2 was designed by using the software BioDraw Ultra 12.0.
- Why authors choose Autodock Vina?
Response: Thanks for the comments and suggestions. The PyRx-Virtual Screening Tool 0.8 is the version of Autodock Vina. The docking results obtained in this program suite were found to be authentic and much similar to the case of Schrodinger drug discovery software.
- The overall analysis and arrangement of this manuscript looks perfect.
Response: Thanks for the comments and suggestions.
- If authors decided to capitalize the compounds names, then it should be maintained throughout the entire manuscript.
Response: Authors are thankful to reviewers for their valuable suggestions. We have revised the compound's name.
- English language need to improve carefully.
Response: Thanks for the comments and suggestions. The English language in the manuscript was thoroughly revised. Spellings and grammatical and typographical corrections were done.
Reviewer 2 Report
The authors are commended for submitting a well-written manuscript investigating several phytochemicals effect on potential target of neurodegenerative pathway in Cerebral malaria (CM). I find this study very interesting to me and sheds light on a very important aspect in potential CM treatment candidates. Recent studies highlight the vital role of the BBB endothelium in CM neuropathogenesis and Molecular Docking on EphA2 receptor could be a vital target. Well written and insightful research with succinct discussion!
Author Response
The authors are commended for submitting a well-written manuscript investigating several phytochemicals effect on potential target of neurodegenerative pathway in Cerebral malaria (CM). I find this study very interesting to me and sheds light on a very important aspect in potential CM treatment candidates. Recent studies highlight the vital role of the BBB endothelium in CM neuropathogenesis and Molecular Docking on EphA2 receptor could be a vital target. Well written and insightful research with succinct discussion!
Response: Thanks for the comments and suggestions. Also, thanks a lot for your acceptance and nice words.
Reviewer 3 Report
The author has reported in the MS "Discovery of Potential blocker of the EphA2 Receptor: Involved in Cerebral Malaria from Tinospora cordifolia, Taraxacum officinale & Rosmarinus officinalis L. Through in-silico Study" which is important in the biomedical field. They have discovered the phytoconstituents from the plant Tinospora cordifolia, Taraxacum officinale & Rosmarinus officinalis L. as the potential blocker of EphA2 receptor for the treatment of cerebral malaria. They have reported 12 phytoconstituents from these plants as the blocker of EphA2 receptor through in-silico docking studies. Some comments are presented below:
Major Revisions:
* The English writing in the manuscript needs thorough revision with correct spellings. It is not up to the mark within this reputed journal.
* The authors should provide discussion and reference regarding relevant studies of EphA2 receptor.
* In the Material and Method section, the author should provide the version of the in-silico toxicity prediction study.
* There is a wide discussion for the toxicity study and ADME study. It should be better if they provided some references and discussed the applied method.
* Provide pictures with clear visibility in Figure 3 and Figure 4.
* Graphical abstract is not clear. It looks hazy. Please submit a clear structure.
* The manuscript should be proofread for typographical errors throughout.
* Conclusion is too elaborative. It should be more relevant to the experimental studies.
Author Response
The author has reported in the MS "Discovery of Potential blocker of the EphA2 Receptor: Involved in Cerebral Malaria from Tinospora cordifolia, Taraxacum officinale & Rosmarinus officinalis L. Through in-silico Study" which is important in the biomedical field. They have discovered the phytoconstituents from the plant Tinospora cordifolia, Taraxacum officinale & Rosmarinus officinalis L. as the potential blocker of EphA2 receptor for the treatment of cerebral malaria. They have reported 12 phytoconstituents from these plants as the blocker of EphA2 receptor through in-silico docking studies. Some comments are presented below:
Major Revisions:
* The English writing in the manuscript needs thorough revision with correct spellings. It is not up to the mark within this reputed journal.
Response: Thanks for the comments and suggestions. The English script in the manuscript was thoroughly revised. Spelling, grammatical and typographical corrections were done.
* The authors should provide discussion and reference regarding relevant studies of EphA2 receptor.
Response: Authors are thankful to reviewers for their valuable suggestions. The studies relevant to the EphA2 receptor are included and discussed in the manuscript with their references.
* In the Material and Method section, the author should provide the version of the in-silico toxicity prediction study.
Response: We thank the reviewer for the comments. These studies were carried out by using open online sources. Hence methodology is not described. But the details of online web source in the form of web link is given in the results and discussion section. These parts are highlighted.
* There is a wide discussion for the toxicity study and ADME study. It should be better if they provided some references and discussed the applied method.
Response: Authors are thankful to reviewers for their valuable suggestions. Some relevant references of the previous studies using the open-source program ProTox-II toxicity explorer and SwissADME have been cited.
* Provide pictures with clear visibility in Figure 3 and Figure 4.
Response: We thank the reviewer for the comments. Pictures were refined for clarity and were included in the manuscript.
* Graphical abstract is not clear. It looks hazy. Please submit a clear structure.
Response: Authors are thankful to reviewers for their valuable suggestions. The clear copy of the Graphical Abstract is now included as a separate PDF file.
* The manuscript should be proofread for typographical errors throughout.
Response: Authors are thankful to reviewers for their valuable suggestions. The manuscript was thoroughly revised and proofread for typographical errors throughout. Necessary corrections were made.
* Conclusion is too elaborative. It should be more relevant to the experimental studies.
Response: We thank the reviewer for the comments. The conclusion part was made concise and more relevant to the experimental studies and findings.

Round 2
Reviewer 1 Report
The authors have done the satisfactory changes. The manuscript can be accepted in the current form.
Reviewer 3 Report
The authors did all the revisions as per my suggestions. The manuscript can be accepted in this current form.